# From blueprint to biobank: Leveraging expert recommendations for implementing change (ERIC) to pediatric cancer biobanking in Pakistan

Javeria Aijaz [1,2]*, Muhammad Rafie Raza[3], Kafeel Naz Sajid[4], Fouzia Naseer[1], Nida Jawaid[2], Saba Jamal[5], Nickhill Bhakta[6], Thomas B. Alexander [7], Megan C. Roberts[8]

**1** Molecular Pathology Section, Clinical Laboratories, Indus Hospital & Health Network, Karachi, Pakistan, **2** Biorepository Section, Clinical Laboratories, Indus Hospital & Health Network, Karachi, Pakistan, **3** Pediatric Hematology/Oncology Department, Indus Hospital & Health Network, Karachi, Pakistan, **4** Electronic Medical Records Department, Indus Hospital & Health Network, Karachi, Pakistan, **5** Clinical Laboratories, Indus Hospital & Health Network, Karachi, Pakistan, **6** Department of Global Pediatric Medicine, St. Jude Children's Research Hospital, Memphis, TN, United States of America, **7** Department of Pediatrics, University of North Carolina, Chapel Hill, NC, United States of America, **8** Division of Implementation Science in Precision Health and Society, University of North Carolina, Chapel Hill, NC, United States of America

* javeria.aijaz@tih.org.pk

## Abstract

### Background

In low- and middle-income countries, limited infrastructure and resources hinder biobank establishment, affecting specimen diversity. Addressing this gap is crucial for equitable health outcomes, as current databases are skewed towards Northern-European populations. In Pakistan, pediatric cancer biobanks are non-existent. Indus Hospital & Health Network (IHHN) in Karachi, with its large pediatric cancer unit, aims to establish a biobank to address region-specific pediatric cancer research needs. This manuscript describes the biobank implementation process using implementation science frameworks.

### Methods

The pediatric cancer biobank at IHHN collects FFPE specimens for solid tumors, and isolated mononuclear cells from peripheral blood and bone marrow of suspected acute leukemia. Implementation planning workgroups included clinicians, EMR, IT, management, senior leadership, IRB, and external support from UNC and St. Jude Children's Cancer Hospital. The selection of applicable ERIC (Expert Recommendations for Implementing Change) strategies through stakeholder workgroups considered scope, budget, and feasibility, and context. Standard protocols from ISBER and BCNet guided alignment with best practices. IHHN's past experiences and tacit knowledge gained through rapid, successful implementation also facilitated strategy

**Data availability statement:** All relevant data are within the paper and its Supporting Information files.

**Funding:** The author(s) received no specific funding for this work.

**Competing interests:** The authors have declared that no competing interests exist.

**Abbreviations:** CAP, College of American Pathologists; DNA, deoxyribonucleic acid; EDMS, Electronic Document Management System; ELSI, Ethical Legal Social Implications; EMR, Electronic Medical Records; EPIS, Exploration Preparation Implementation Sustainment; ERIC, Expert Recommendations for Implementing Change; FFPE, Formalin Fixed Paraffin Embedded; HMIS, Hospital Management Information System; IHHN, Indus Hospital & Health Network; ISO, International Standards Organization; LMIC, low- and middle-income countries; PBMC, peripheral blood mononuclear cells; RNA, ribonucleic acid; SOP, standard operating procedures.

selection. The EPIS framework (exploration, preparation, implementation, sustainment) was used to map and organize the selected intervention strategies.

## Results

Biobank implementation at IHHN, organized by EPIS stages, has been described through a set of 41 implementation strategies. Of these, 34 were selected out of 73 originally published ERIC strategies, while 7 were added based on contextually based workgroup consensus. 599 acute leukemia and 1137 solid tumor specimens have been banked since inception of the biobank operations 2 years earlier. The implementation activities and challenges described include infrastructure, swift specimen collection, prior to treatment, and informed consent. The ancillary processes including training and quality control have also been described and related data presented.

## Conclusion

The implementation of Pakistan's first acute leukemia biobank using ERIC and EPIS frameworks offers a structured approach beneficial for settings with limited biobanking experience. This intervention aligns with recognized implementation science frameworks, while addressing aspects pertinent in low- and middle-income countries.

## Background

The core function of a biobank, as outlined by the European Commission, is the ongoing collection and retention of biological specimens with associated medical data for potential research projects. The function must be executed with in-built mechanisms for donor privacy, along with a secure mechanism for re-identification to allow sharing of clinically significant data with the patient, if needed. Additionally, well-defined governance structures and procedures that uphold the rights of specimen donors, and the interests of other stakeholders, must form integral components of all processes [1]. In low- and middle-income countries, limited infrastructure and constrained resources pose challenges to establishing and maintaining biobanks, hampering the inclusion of diverse and representative specimens for research. Data from these specimens feeds into medical knowledge platforms for designing diagnostic and treatment interventions. Addressing this gap is imperative to ensure equitable health outcomes across populations, as current databases, underpinning critical diagnostic and treatment designs, are heavily skewed towards Northern-European populations [2].

In Pakistan, a low-middle income country of 240 million inhabitants, pediatric malignancies account for around 8–10% of total cancer cases [3,4]. Leukemia is the most common pediatric cancer, followed by brain tumors and lymphomas. Pediatric cancer biobanks, specifically dedicated to systematically gathering, preserving, and

studying biological samples of pediatric leukemias are non-existent in the country. Maintaining such biobanks, along with utilization of these specimens for research, could reduce region-specific knowledge gap of pediatric cancer characteristics, and tailored diagnostic and treatment approaches. Indus Hospital & Health Network (IHHN), Karachi, has one of the largest pediatric cancer units in Pakistan, with around 1200 new diagnoses and patient registrations annually. Additionally, in alignment with its overarching mission, the hospital provides quality cancer care at no charge to the patient [5]. Given its significant patient base and expansive network, the hospital is optimally positioned to establish a biobank that encompasses the diversity of pediatric cancer cases in the country. This manuscript describes our implementation process of a pediatric cancer biobank at IHHN, Karachi.

Process frameworks such as Exploration, Planning, Implementation, Sustainment (EPIS) [6] can guide the implementation of an evidence-based program, such as a cancer biobank, while published change strategies such as Expert Recommendations for Implementing Change (ERIC) [7] can inform execution blueprints through systematic inclusion of a spectrum of cross-disciplinary implementation strategies. Selected ERIC strategies, aligned with the successive EPIS stages, were thus used to develop an action plan to implement the intervention. The approach, utilizing the published, and widely known EPIS-ERIC syntax, can advance our understanding of the most impactful implementation approaches, especially for comparable interventions and settings, i.e., countries which are similarly constrained in terms of infrastructure, processes, and trained human resource.

## Methods

### Description of intervention

The pediatric cancer biobank at IHHN was designed to collect peripheral blood and/ or bone marrows of suspected acute leukemia patients prior to the start of therapy, isolation of mononuclear cells from these specimens, and their storage at appropriate temperatures (Fig 1). In addition, the bank also comprises Formalin Fixed Paraffin Embedded (FFPE) specimens for solid tumors. This intervention, centered in the clinical laboratory, was implemented through a team of clinicians, electronic medical records (EMR) department, IT department, managers, senior leadership, and IRB. In addition, there was external technical support from University of North Carolina (UNC) and St. Jude Children's Cancer Hospital, USA. Sample collection started on 1st June 2022, following IRB approval (IHHN_IRB_2021_12_019).

### Selection of ERIC strategies

ERIC are a compilation of 73 strategies, grouped into 9 major content areas, covering multiple facets of an intervention [8], though all strategies may not be applicable under all circumstances. Applicable ERIC strategies were selected using stakeholder workgroups, including in-person and online meetings and discussions. Exclusion criteria included mismatch with the scope of the intervention, budget constraints and feasibility. For example, the biobank, was initiated at a single institution, rendering a majority of strategies designed for large, multi-center programs inapplicable. Budget constraints precluded immediate adoption of some other strategies, e.g., using capitated payments. Standard protocols, available through open sources and organizations like ISBER and BCNet, were also consulted to ensure that the implementation effort is in alignment with standard practices. This assisted in modification of some strategies to fit the context. For example, since biobanking accreditation program is offered by international organizations, e.g., College of American Pathologists (CAP), International Standards Organization (ISO), the ERIC strategy of changing accreditation requirements was, as such, not considered applicable. Rather, a modification to 'pursuing accreditation' in the light of context was considered more appropriate.

Notably, strategy selection process built on past implementation experiences of IHHN in general, marked by rapid, successful implementation of many strategies and interventions. At the organizational level, these include expansion from a single tertiary care hospital to a country-wide healthcare network in a matter of 15 years [5]. The clinical laboratory expanded from a small facility less than a decade ago to a highly specialized laboratory referral laboratory for the

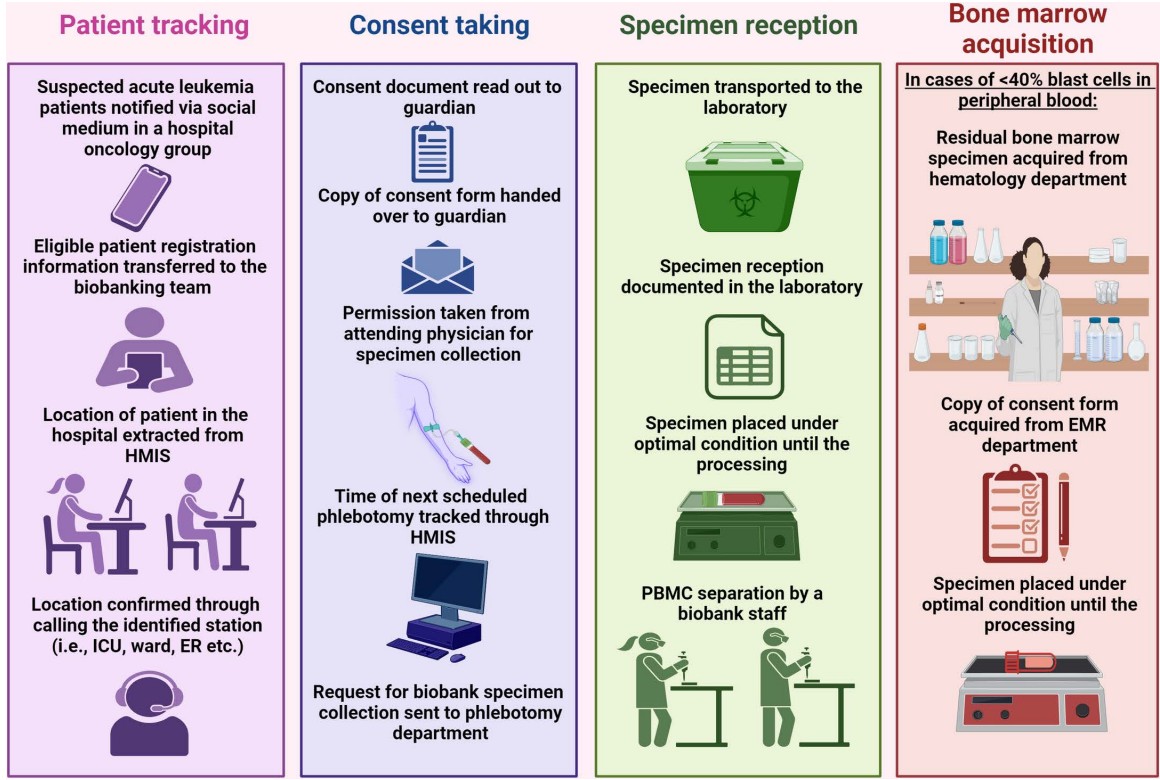

**Fig 1. Patient-tracking, consent-taking, sample reception, and bone marrow acquisition steps involved in acute leukemia biobanking at IHHN, Karachi.**

network, accredited with College of American Pathologists (CAP) [9]. The tacit knowledge thus, at the institutional and technical level, assisted with the selection of strategies through workgroups, accounting for consideration of some strategies extraneous to originally published ERIC as well. It was recognized though that the initial strategies outlined may need to be adapted through the unique challenges, both anticipated and unanticipated, encountered during the actual implementation.

## Implementation science framework

While most implementation processes can be broadly divided into stages of progress such as exploration, preparation, implementation and sustainment (EPIS), it is recognized that the timing of the precise actions under these stages do not always follow a chronological order. Thus, while a demarcation of EPIS into a neat sequence of events is impractical, a mapping of the ERIC strategies and actions undertaken under a process model such as EPIS provides a broad organizing framework for the intervention and implementation activities.

## Results

To date (June 2024), 599 acute leukemia and 1137 solid tumor specimens have been banked since inception of the biobank operations 2 years earlier (Fig 2). The total number of new patient registrations for acute leukemia is in Table 1, along with the percentage of samples banked for each period. Some reasons for missed collections include physician advice against additional specimen collection, patients opting out of treatment and leaving, initiation of therapy before the specimen could be drawn, and lack of patient consent.

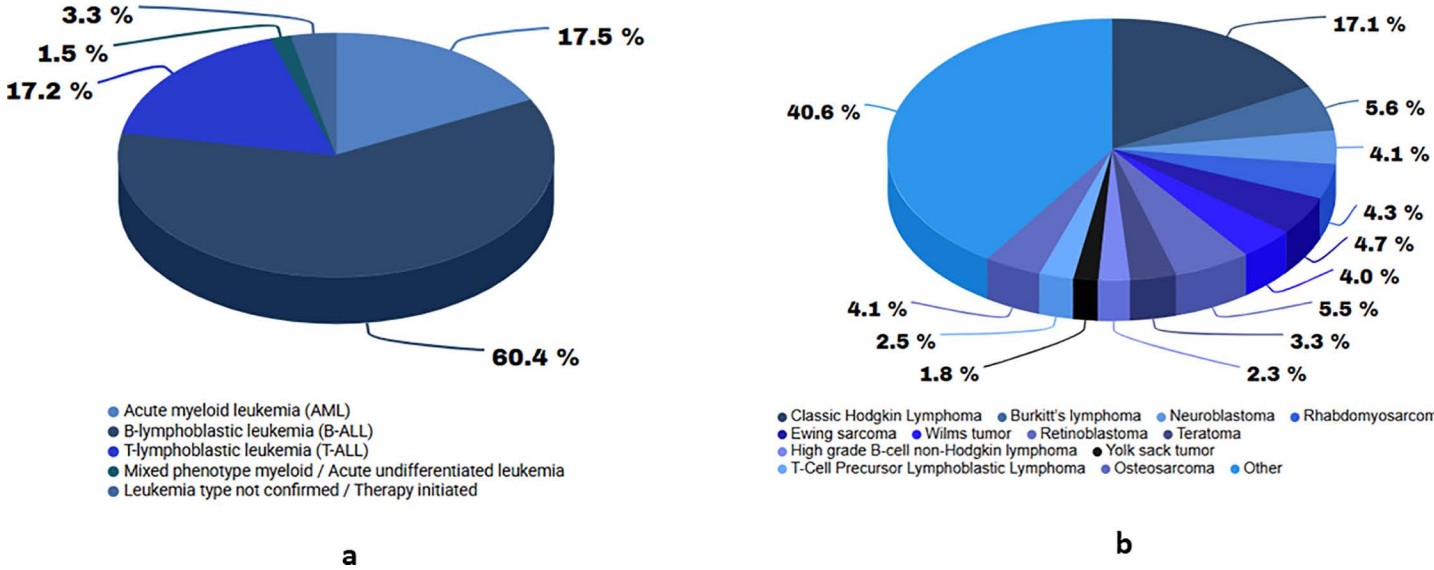

**Fig 2. Distribution of biobank PBMC (a) FFPE (b) and specimens by primary diagnosis.**

**Table 1. Acute leukemia specimen banking: New patient registrations versus the number and percentage of specimens banked.**

| Acute leukemia banking as a percentage of new registrations | | | |
|---|---|---|---|
| Time Period | New registrations | Samples banked | % of banked samples |
| June to Dec 2022 | 260 | 201 | 77% |
| Jan to Dec 2023 | 350 | 232 | 66% |
| Jan to June 2024 | 200 | 166 | 83% |
| **Total** | **810** | **599** | **74%** |

The succeeding paragraphs, organized by EPIS stages, describe the specific actions or ERIC strategies (indicated in bold font), as applied to establishing a pediatric cancer biobank at IHHN. The selected ERIC strategies, as aligned with the EPIS framework described below, are shown in **Fig 3**.

### Exploration

A **sense of urgency** for installation of the biobank was facilitated through a research collaboration with St. Jude Children's Cancer Hospital, and University of North Carolina (UNC), USA. Identification of potential near-term research collaborations up front to utilize the banked samples assisted in making the case to the executive boards for facility upgradation and equipment procurement, while also offsetting some reagent costs. It was formalized with submission to the board, and approval, of a 'need statement' focusing on 4 principal advantages (research, diagnostic assay validation, diagnostic assay development, revenue generation) of an organized system for storing and cataloging patient specimens at IHHN, and the creation of a separate biobanking unit within the clinical laboratory, respectively.

**Involvement of executive boards** (Executive Medical Directorate) at an early stage helped ensure that the **change is mandated** across multiple stakeholders. We used workgroups with these stakeholders (clinicians, electronic medical records (EMR) department, IT, managers, senior leadership, and IRB) to **build a coalition** providing input and advice on developing an initial scope, and an ERIC-based **implementation blueprint**. In addition, we **assessed local needs** and **assessed barriers and facilitators** for implementation of the biobank through the same process, which simultaneously

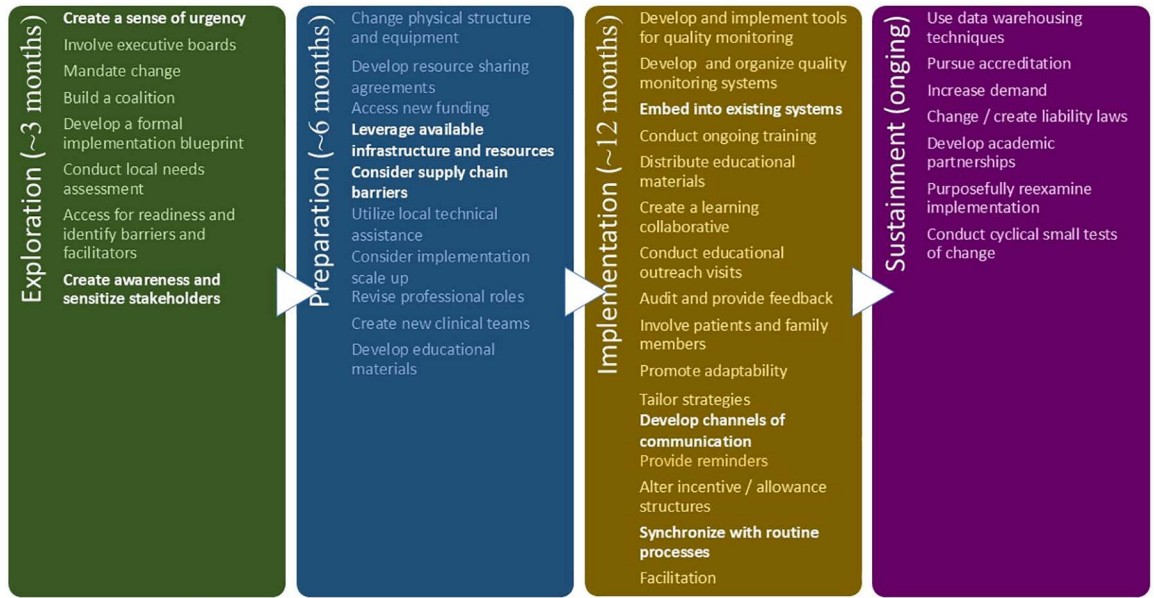

**Fig 3. ERIC strategies mapped to EPIS process framework, as applied to the pediatric cancer biobank at Indus Hospital & Health Network (IHHN).** Strategies not listed as such in the original work but found useful in our implementation experience have been highlighted.

helped **create awareness, and sensitization**, while addressing concerns and developing plans for overcoming particular barriers. Some of the early barriers identified in these meetings were lack of prior experience in the country with acute leukemia biobanks, and lack of specific and dedicated space and training at the institution. Facilitators included the research partnership to utilize samples, a quality systems accredited pathology laboratory, commitment by senior leadership, as well as extensive prior experience with implementation of other new interventions at the organizational and technical level.

### Preparation

**Changes in physical structure and equipment** were needed to allow the safe storage of liquid nitrogen, as the unit was not included original design plan of the clinical laboratory at IHHN. A 6'-0" x 16'-0" enclave, sufficient for around 10,000 cryovials in liquid nitrogen storage tanks was carved within a larger room, ultimately compromising one of the meeting rooms. This was compensated through a **resource-sharing agreement** with another department.

While the research collaboration offset some of the reagent cost, infrastructure alterations required **access to new funding**, organized through a dedicated philanthropic donation by the executive medical director's office. New acquisition was minimized through **leveraging existing infrastructure and resources** (S1 Table) e.g., large equipment (freezers, biosafety cabinets, centrifuges etc.,) space, personnel. FFPE banking was initiated without additional facilities, requiring only supplementary processes, e.g., consent and data annotation to the biobank. For organizations lacking liquid nitrogen storage infrastructure, the –80°C freezers, available in most clinical laboratories, can be used to store isolated PBMC genomic DNA or RNA in an appropriate long term storage medium (e.g., Zymo RNA shield). These minimum investments could enable low- and middle-income countries to have available specimens for use in research, while enriching population-wide databases used to develop therapies and diagnostics.

Based on prior laboratory experience, it was known that some specialized reagents and equipment are sporadically available due to lower demands and elaborate import processes. Therefore, ordering of equipment and facility upgrade items were prioritized, while reagent procurement orders were made upon equipment delivery, or with a confirmed delivery date to avoid expiry and wastage. The lag time was utilized for other, concurrent implementation measures. **Considering**

**supply chain challenges** early in the implementation process is crucial in setting similar to ours where these can critically hinder smooth execution.

**Local technical assistance** was needed for the design of liquid nitrogen storage facility, i.e., special safety considerations. In particular, the most feasible space identified could not directly exhaust, requiring that an exhaust line be installed atop the adjacent BSL-III laboratory. One of the IHHN new building project team members with prior expertise in the design of liquid nitrogen storage facilities in another country, was identified by the senior management to guide the process. Challenges with installation of the requisite equipment included the local non-availability of some items (oxygen sensors), while the liquid nitrogen containers available locally did not have in-built temperature monitoring devices and alert systems. The organizations biomedical engineering department identified alternate system of wireless temperature monitors and data loggers to be procured as accessories, and integrated with the existing wireless temperature monitoring and alarm system of laboratory.

The ad hoc use of available infrastructure, however, will become impractical later in the biobank's expansion trajectory. With this recognition, simultaneous **scale-up provisions** were made for an enlarged biorepository within the upcoming IHHN building, presently in the construction phase and slated for completion in approximately two years. With projections of ongoing biobank growth, the new space is allocated to accommodate a minimum of 100,000 cryovials within liquid nitrogen storage tanks, to be funded together with the new building by philanthropic donations, government, and others. The integration of the biobank within the hospital infrastructure, and its recurring budget into that of the pathology department, will support its longer-term sustainability, while specimen utilization in research projects is anticipated to similarly offset some operational costs.

**Revision of professional roles** was undertaken to initiate biobank operations as a separate unit within the clinical laboratory, albeit with hiring of no additional staff. Instead, **new clinical teams were created** through assignment of one of the existing pathology section heads to supervise the biobank, with such supervision rotating after period of 3 years, while other relevant procedures were distributed within the clinical, phlebotomy, and pathology departments. Corresponding changes in documents, e.g., organograms, job descriptions, was accomplished in liaison with the human resource department. Although there was an agreement that the current molecular pathology's technical staff will perform the additional biobanking assignments, it was recognized by the hospital management that HR requirements will be reconsidered once the scope expands, and the current arrangement fail to fulfill the biobank's needs.

Standard Operating Procedures served as **educational materials** for staff training, together with a grant from the International Society for Biological and Environmental Repositories (ISBER) allowing a laboratory technical staff member to acquire certification in biobanking by the American Society of Clinical Pathology (ASCP). The certification, together with templates provided by UNC and open sources, helped incorporate best practices of biobanking into the developed protocols. The protocols were also pre-tested, and amended, where needed, before implementation. For example, the template SOP-recommended DPBS-Phosphate buffer was expensive and not readily available. RPMI-160, a commonly used item in IHHN's cytogenetics laboratory, and more cost-effective, was evaluated and substituted through a comparison of PBMC isolation counts and viabilities. The optimum rotation speed for whole blood specimens before PBMC isolation was also empirically determined to avoid hemolysis. Similarly, the recommended whole blood dilution ratio before addition to Ficoll rendered red-tinged isolated mononuclear cells in specimens with high total leukocyte counts. The dilutions were ultimately adjusted to vary with total leukocyte count of the specimens. Following these adjustments, the initial validation parameters used gave optimal results, including flow cytometrically-determined comparable expression of surface markers (S2 Table, S1 Fig).

## Implementation

Tools for quality monitoring include physical appearance of isolated PBMC (e.g., color), cell viability following PBMC isolation through trypan blue staining, and quality and quantity of extracted nucleic acid (Fig 4, S1 File). Budget constraints were, however, prohibitive to ascertainment of other, more sophisticated, quality parameters such as viability of PBMC cell

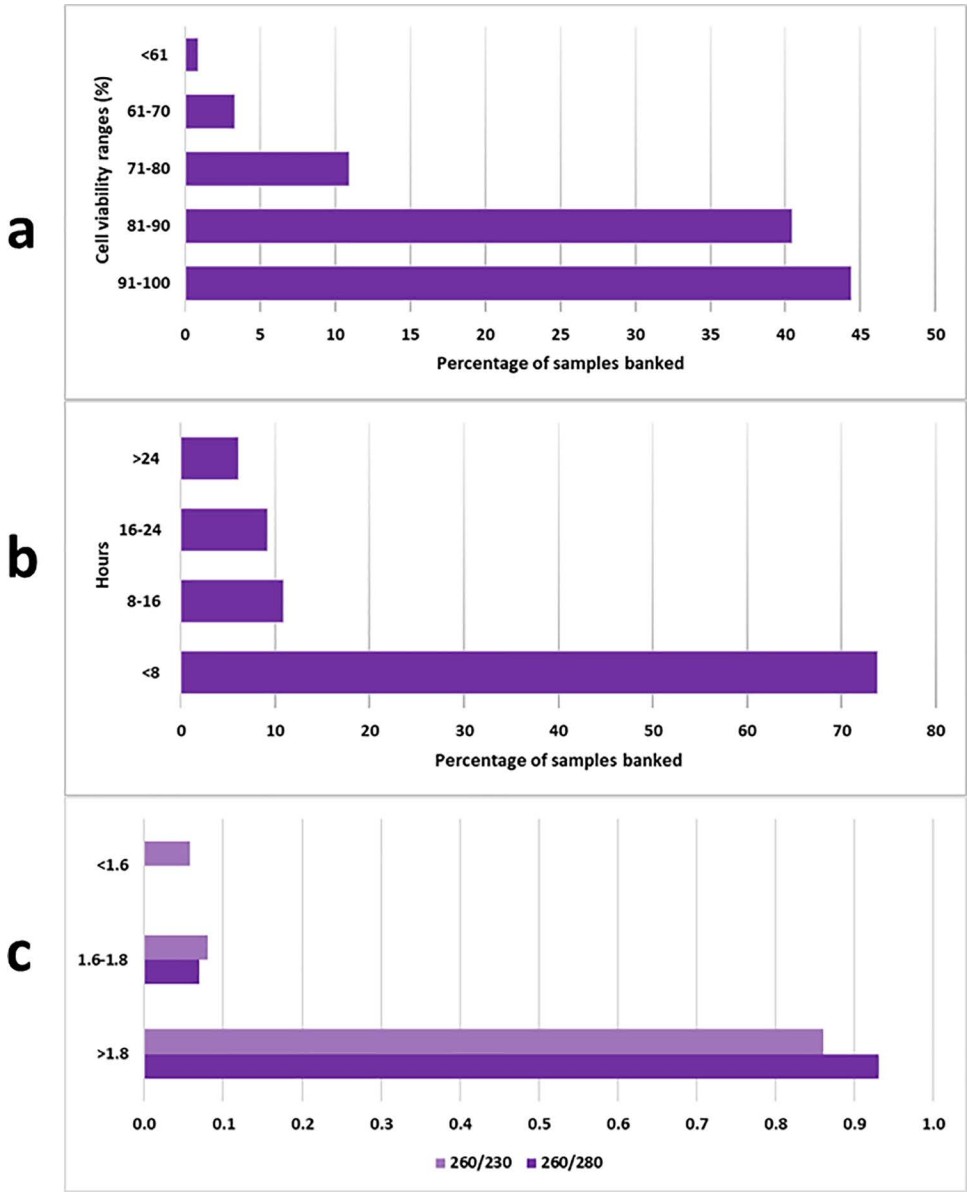

**Fig 4. Quality metrics of PBMC specimens at IHHN, Karachi.** Collection to storage intervals (a) are less than 24 hours for most specimens at present, which is an optimal interval reported by many studies for most functions, though the ideal time reported by others is up to 8 hours following collection. Currently, specimens received before 5 pm are being stored within 8 hours of collection, while this limit is exceeded with specimens arriving after 5 pm as these need to be processed the next day. Collection-storage intervals exceeding 24 hours have mostly been for initial specimens when sample flow, and other protocols were less well defined. Trypan blue viability prior to storage (b) is above 90% for most specimens, but a range of pre-storage viabilities are nevertheless observed. All specimens are stored as these may have different potential uses. In our experience, cells with up to 70% pre-storage viability give optimum RNA yield and quality for transcriptome sequencing, while it is possible that for more sensitive uses like cell culture, specimens with higher viability will need to be selected. Spectrophotometrically determined wavelength ratios (c) for extracted RNA from specimens shows 260/280 ratio of 1.8-2, and 260/230 of more than 2 for most specimens.

cultures. Sample tracking through barcoded labels, in a manner similar manner to diagnostic specimens, allows accurate monitoring of timelines between the various collection to storage steps, while storage temperatures are continuously monitored through automated data loggers and alarm systems.

The biobank at Indus Hospital & Health Network, Karachi, is embedded within a College of American Pathologists (CAP) accredited laboratory with a **developed and organized quality management system**, which was applicable to the biobank as well. This includes general facilities management, e.g., monitoring and maintenance of environment, and equipment; contingency and safety plans such as oxygen sensors and alarm systems, liquid nitrogen level monitors, and backup power supplies; requirements related to staff, e.g., training and competence assessment, and others. In this respect, **embedding a biobank** within an established clinical laboratory in the initial stages was of advantage, allowing for the infrastructure, material, and human resources to be shared, while simplifying initial implementation of the overlapping quality management system, by staff proficient in these principles and practices.

Additionally, evaluation of the performance indicators, feedback, and discussions on quality improvement, quality control failures, root cause analysis, corrective and preventive actions has been incorporated as components of regular departmental meetings and management review committee meetings. The latter are also used to **audit and provide feedback** on the quality monitoring data, while regular internal audits in a manner similar to that for clinical laboratory are used as opportunities to identify any gaps against recommended standards, and to take corrective and preventive actions.

**Ongoing training** was needed regardless, for the core processing of samples on account of a shortage staff dedicated to biobanking tasks, as well as for ancillary processes. For example, the few staff trained initially were unable to manage all eligible collections, which required continual vigilance and ongoing communication with phlebotomists, porters and other attending medical staff for collection and transportation. Over time, however, all staff in the molecular pathology section were trained allowing more redundancies and availability of trained staff. **Distribution of educational materials** (SOPs, tools, e.g., competence assessment, worksheets) through the institutions Electronic Document Management System (EDMS) ensures updated access to all staff.

At the same time, the biobank's free of charge membership with the Biobank and Cohort Building Network (BCNET) [10], as well as ISBER [11], provided for a for a **learning collaborative** which assists with ongoing education, training, and quality improvement, e.g., through the ISBER's open forum for discussions. **Outreach educational visit** to an institution in Pakistan with a solid tumor biobank, also a member of BCNet [12], have also been organized to facilitate mutual learning and collaboration.

**Involving patients and family members**, within the context of biobanking, was interpreted as applying the relevant elements of the Common Rule [13], pertaining to broad consent. The institution's ethical committee approved the protocol and consent form, in English (the official language of Pakistan) and Urdu (the local language of Pakistan) (IHHN_IRB_2021_12_019). A template broad consent form provided by UNC was modified, where needed, to enhance comprehension and include local process flows, through incorporation of clinicians' and ethical review board's input, and pre-testing. For patients sufficiently mature to understand PBMC specimen collection, banking, and utilization protocols, verbal assent is solicited. However, the form is signed by a parent or legal guardian, with a copy provided to them. As regards solid tumor banking, a broad consent for use of residual specimens in research is routinely sought at the time of the relevant procedure and retained with the EMR. FFPE specimens from consenting patients are filtered, and relevant associated data added regularly to the biobank. To date, informed consent has not been received from 13 of the eligible acute leukemia and 296 solid tumor patients.

While broad consent allows flexibility in pursuing science, it necessitates that the patient understand and agree with not receiving any specific details of subsequent studies using their samples [14,15]. As broad consent is recognized as the only feasible solution for biobanking currently, there are international guiding frameworks for consent document contents. ESLI biobanking research, and national guidelines which can potentially consider the specific cultural and social milieu of Pakistan for designing of broad consent forms, other policies, and governance of biobanking are, however, entirely lacking. Thus, while the implemented consent form is based on an international template, the **need for adaptations** against the local context, as such guidance becomes available, is fully appreciated. As one example, Pakistan is a multilingual

country with a substantial proportion of patients speaking only their native language, creating language barriers. Translating consent forms into several Pakistani languages does not fully address the issue as the overall literacy rate in Pakistan currently stands at 60%, while several native languages do not have a written script. In such situations, on-call attending medical staff, or other attendants, who speak and understand the particular native language are asked for assistance in communication between biobank staff and patients or their guardians.

The Belmont Report's Principle of respect for persons requires particular consideration in these circumstances given the vulnerable populations involved, including children and individuals not formally educated. Ethical dilemmas can potentially arise with more than minimal risk research, given the unresolved debates on whether a large societal benefit can outweigh individual risk, and if such research should proceed at all. At the same time the principle of justice also demands equitable distribution of not just risks but also benefits. While sample selection is based purely on technical eligibility rather than social characteristics, the inclusion of vulnerable populations is necessary for developing targeted treatments that will benefit these communities. To minimize any perceived coercion, consent is obtained by independent personnel unrelated to treatment. Staff clearly communicate that treatment decisions are entirely independent of participation. Notwithstanding, some degree of uncertainty about the completeness of relevant information transferred and comprehended currently remains inevitable for our patient population. Future work in this direction may entail extensive, close work with relevant communities through identification of individuals proficient in English, Urdu, and the local language, together with the right level of education and cultural understanding, to design non-written (e.g., audio) supplementation of consent form information.

Protocol safeguards for biological sample collection establish strict protective measures for donors. These include prohibiting monetary compensation to prevent exploitation and mandatory physician approval for collection to ensure samples are not drawn when it may not be absolutely safe for the patient. Collections are aligned with diagnostic draws to maintain standard safety protocols. While physical risks from sample collection are minimal, others including psychological, social, legal, and economic, are minimized, though not entirely eliminated through de-identification, and other measures to protect privacy and confidentiality.

The initial implementation blueprint included physician-initiated consent-taking and specimen collection as a first stage of the process map. This was deemed necessary for the prompt action needed for PBMC specimen collection and banking. However, the **strategy needed to be tailored** soon as only a small percentage of eligible collections were possible with the strategy despite several attempts at improvement. Based on this experience, dedicated staff from the clinical laboratory were trained and assigned this role, which increased collections, and segregated consent-taking staff from those administering treatment. This necessitated continued vigilance, however, due to the absence of a system to promptly alert biobank staff of eligible patient registrations. With the adapted specimen collection strategy, however, traditional means of communication (e.g., email, landline phone) did not allow coordinated execution of the requisite steps (e.g., informed consent, locating patients) prior to specimen collection in a timely manner. More sophisticated development of information systems for this would have been time-consuming and requisite of more initial budget. To circumvent the issue, WhatsApp groups of the hospital oncology team for new patient registration notification were utilized as conduits for eligible patient notifications to biobanking staff. This process, however, is also not always smooth, especially as patients are at times shifted from one unit to another (e.g., ER to ward) physically, but the electronic Hospital Management Information System (HMIS) reflects transfer subsequent to other hospital transfer protocols, necessitating development of continuous **channels of communication** with several departments.

As IHHN is primarily a health service delivery institution, biobank collections were at times not prioritized, requiring **ongoing reminders** to inculcate the practice of specimen collection for biobanking. Later, biobank requisitions were added to the routine HMIS, allowing biobank staff to electronically generate requests for collections. Any missed collections are thus now reflected in the electronic system, **disincentivizing** missed collections, which are one of the phlebotomy staff's key performance indicators as well.

Taken together, **synchronization with routine processes** worked to increase eligible collections manifold, through leveraging the routine workflow (i.e., WhatsApp group) of physician-generated new eligible patient notifications; integration of specimen collection requisitions into the routine hospital HMIS; and synchronizing collections with routine laboratory diagnostic blood draws. The latter also **facilitated** consent in comparison with situations requiring separate blood draws for diagnostic and biobanking purposes.

## Sustainment

Information technology is vital to quality systems in biobanks but needs to be **tailor-made** once workflows have been solidified, and experience gained in the unique institutional requirements. An in-house Biobank Information System (BIS) enhancing the overall functionality and effectiveness of the biobank is under development at IHHN. Nonetheless, challenges related to data storage, annotation, processing, and retrieval (or **data warehousing**), with all of the ethical and data confidentiality requirements of biobanking will be essential and addressed on a continual basis, with availability of more funding. To enable international data sharing and collaborations, a process of developing material and data transfer agreements, compliant with the legal frameworks and ethical considerations is in place. Currently, data sharing is done following ethical and legal approval, as well as through utilizing secure cloud-based platforms which meet international data security standards for data transfer. As the biobank expands, standardized data formats and protocols compatible with major international biobank networks will need to be developed. Moreover, the consent process necessitates prior approval of deidentified data sharing.

Enrolling in a proficiency testing program, and **pursuing accreditation** will also need to from part of the continual quality improvement initiatives. Although IHHN's clinical laboratory currently has accreditation with College of American Pathologists (CAP), the organization does not offer an international program for biorepositories, while ISO accreditation, implemented by Pakistan National Accreditation Council (PNAC) in Pakistan, is not available for ISO 20387:2018. Creation of more biobanks in the country will **increase the demand** necessary to initiate such accreditation process of biobanks in the country, in a manner similar to that of clinical laboratories in Pakistan [16]. A governance framework, including **liability laws**, for ethical handling of specimens and associated data will also need to be created. Initiation of biobanking at other centers will create the impetus for promulgating laws providing ethical mandates, under which individual institutional policies can be developed.

Other ongoing sustainment efforts include **fostering additional academic and non-academic partnerships** with local and international researchers, clinicians, and institutions interested in utilizing the biobank's resources. Some future initiatives include the expansion of scope of the biobank through the establishment of collection and storage mechanisms for frozen tissues. Although FFPE specimens can be useful resources, the quality of extracted nucleic acid (especially RNA) often falls short of requirements. Similarly, expanding the scope of the PBMC specimen collection to include treatment follow-up specimens would be another direction of the expansion. **Scale-up** of scope to other network facilities, or even beyond to other healthcare and research facilities, however, will require uniformity of protocol implementation at other sites, effective coordination and communication mechanisms, and infrastructure for timely transport of specimens. This can be relatively straightforward for stable specimens, and those without stringent temperature control requirements. While IHHN operates as a network with partial centralization of laboratory services, specimens for PBMC isolation and storage necessitate more stringent monitoring compared to routine clinical laboratory samples, and thus critically dependent on reliable, fast, and feasible transport infrastructure across longer distances. Additional challenges include effective education of participants in semi-urban or rural areas.

## Discussion

A multi-disciplinary team at IHHN identified strategies across implementation phases to plan and establish a pediatric cancer biobank at the institution. Sustainment and scaling up efforts will continue to be conducted as **small cyclical tests of**

**change,** together with **purposefully reexamining implementation**. The ERIC and EPIS frameworks provided structure to planning and implementation, while some specific actions may be unique to our setting. Broadly, strategies and actions also aligned with the barriers and facilitators – factors more generalizable across similar settings in terms of geographic location (i.e., Pakistan) or income level (i.e., low- and middle-income countries). To our knowledge, the biobank at IHHN is the first acute leukemia biobank in Pakistan, while a solid tumor biobank has also been established at Shaukat Khanum Memorial Cancer Hospital & Research Center, Lahore, Pakistan [17].

EPIS, as an implementation science process framework, has mostly been studied in HIV [18–20], and mental health [21–24] interventions, while ERIC strategies have also predominantly been applied to healthcare settings focusing on specific patient groups [25–28]. Generally, in these studies, selected patient groups form the direct target of the intervention. In contrast, though biorepositories do not directly influence the health outcomes of the population donating their specimens, these facilitate improve health outcomes of the population on the whole through research incorporating characteristics of the given population, development of newer diagnostics, and treatments. This manuscript thus offers an implementation science perspective to the intervention of biobanking, an approach not extensively explored or documented until now. Although experiences pertaining to biobanking have been previously published [29,30], an implementation science perspective offers unique advantages to such reports. The constructs, themes, and strategies are designed to be broadly generalizable, and hence applicable to many settings. This may facilitate planning of biobanking interventions through providing structure, common language, and insight into the broad strategies and contexts which are useful or otherwise for the intervention.

Although implementation science is a relatively newer field in its own right [31], it incorporates diverse frameworks, application of frameworks, study methodologies, and contextual backgrounds. While both EPIS and ERIC have been widely applied, the mapping of both ERIC to EPIS has not been documented in any published study. The advantage of combining this approach, for our particular purpose, was to orient the reports direction towards the specifics of strategies and actions. The specific action or activity-level description of this paper may particularly be useful for organizations lacking prior experience and technical expertise in biobanking. Additionally, the description in this manuscript is from an insider's frame of reference, and meant to cover a level of detail not assumed to be possible with studies conducted in the form of assessment of barriers and facilitators, or survey findings conducted by individuals external to the organization.

Lovero et al [32] have conducted a thorough review on the utilization of ERIC strategies in LMIC. Although ERIC is the most commonly used compilation of strategies in Implementation Science, with over 3000 citations, only 60 of these have altogether been reported from low- and middle-income countries until March 2023. Even in these studies, the median number of strategies reported was 6, while the maximum was 46. Significantly, the authors note that some strategies critical to implementation in LMIC, such as tailoring strategies, capturing and sharing local knowledge, utilizing local technical assistance, visiting other sites, and starting a disseminating organization were seldom used. Strategies addressing system level changes including changing liability laws, and preparing patients/consumers to be active participants, were almost never used. Others having more overt budget implications or applications to private healthcare systems, i.e., altering patient/consumer fees, developing disincentives, making billing easier, using capitated payments, and other payment schemes were also rarely used. In comparison, the present work reports on 31 ERIC strategies, while including some of the less frequently reported in LMIC (e.g., tailoring strategies, visiting other sites). The work also includes 7 strategies not explicitly forming a part of the published canon of 73 ERIC strategies. It is recognized though that the depth of strategy utilization and its complexity may vary by a large margin. Thus, ours was a single center implementation experience, reported soon after initiation. With more complex interventions, and increasing temporal spans, the penetration of most strategies in terms of specific actions encompassed will undoubtedly increase.

Some limitations of our report include a lack of formal testing of strategy effectiveness, grounded in an established strategy assessment process, or formal evaluation by an external body. Similarly, an implementation outcome remained undefined in concrete terms. In the case of healthcare settings, the outcome of the intervention is patient-centered, e.g.,

improved utilization of the intervention, improved health metric targeted by the intervention. In our scenario, where the desired outcome is essentially storing specimens in a quality-controlled manner, and with the ethical requirements in place, implementation outcome would be best ascertained by a government licensing body or an international quality systems accreditation body like CAP and ISO. While a government licensing body for clinical laboratories, hospitals, and other healthcare settings exists in Pakistan [33], some organizations opt for more stringent standards through voluntary accreditation by organizations like CAP and ISO. The effectiveness of our implementation would also ideally be gauged by a voluntary accreditation organization. Additionally, as biobanking takes hold in Pakistan, more policy and organizational level strategies and actions will need to be applied for the implementation outcome to extend beyond the single center.

## Conclusion

Implementation of Pakistan's first acute leukemia biobank using a structured approach of ERIC and EPIS frameworks may be especially beneficial for situations where biobanking experience is limited. Future efforts should focus on expanding biobank communities of practice in low- and middle-income countries through experience-sharing of structured implementation science approaches. Further development of physical infrastructure, optimization of existing resources, and implementation of targeted training programs will be indispensable for further development and sustainability of pediatric cancer biobanking in Pakistan, as will be implementation of system-level strategies frequently overlooked in low- and middle-income countries, including addressing regulatory environments and community engagement.

## Supporting information

**S1 Table. List if items available and procured, respectively, for pediatric acute leukemia biobanking at Indus Hospital & Health Network.**
(DOCX)

**S2 Table. Post-thaw PBMC viability and surface marker expression tested through flow cytometric analysis.**
(DOCX)

**S1 Fig. Post-thaw PBMC mean fluorescent intensity, tested through flow cytometry in comparison with those obtained with fresh whole blood, at the time of diagnostic flow cytometry.**
(TIF)

**S1 File. PBMC quality parameters.**
(XLSX)

## Acknowledgments

This work would not have been possible without the unwavering support of the St. Jude Global and University of North Carolina USA. We also extend our deepest gratitude to the dedicated teams of biobank staff, clinicians, laboratory staff, phlebotomy staff, electronic medical records department, and senior management at IHHN whose consistent efforts were instrumental for this endeavor. Special thanks to Dr. Muhammad Shamvil Ashraf, Executive Medical Director for his ultimate leadership in establishing a free of charge pediatric cancer unit at IHHN; Dr. Nazia Khursheed and Maqboola Dojki for their support in establishment of the biobank; Dr. Omer Javed and Talha Israr for their assistance in banking of residual bone marrow samples; Shakir Hussain, Jibran Khan Shariq Ali, Dheeraj Kumar, Ayesha Fatima, and Muhammad Rashid for their invaluable assistance in PBMC specimen collection, handling, and processing.

Finally, this manuscript is dedicated to the brave and resilient children and their families and caregivers who entrusted us with their precious samples for the creation of the biobank. This trust, and conviction in the power of science to transform lives, has been the cornerstone of our progress.

## Author contributions

**Conceptualization:** Javeria Aijaz, Saba Jamal, Nickhill Bhakta, Thomas B. Alexander, Megan C. Roberts.

**Data curation:** Javeria Aijaz, Kafeel Naz Sajid.

**Formal analysis:** Javeria Aijaz, Fouzia Naseer, Thomas B. Alexander, Megan C. Roberts.

**Funding acquisition:** Saba Jamal.

**Investigation:** Javeria Aijaz.

**Methodology:** Javeria Aijaz, Thomas B. Alexander, Megan C. Roberts.

**Project administration:** Javeria Aijaz, Muhammad Rafie Raza, Fouzia Naseer, Nida Jawaid.

**Resources:** Saba Jamal, Thomas B. Alexander.

**Supervision:** Javeria Aijaz, Saba Jamal.

**Visualization:** Javeria Aijaz.

**Writing – original draft:** Javeria Aijaz.

**Writing – review & editing:** Nickhill Bhakta, Thomas B. Alexander, Megan C. Roberts.

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
