## [Decision Letter · Decision Letter 0]

29 Jan 2025

PONE-D-24-44873From Blueprint to Biobank: Leveraging Expert Recommendations for Implementing Change (ERIC) to Pediatric Cancer Biobanking in PakistanPLOS ONE

Dear Dr. Aijaz,

Thank you for submitting your manuscript to PLOS ONE. After careful consideration, we feel that it has merit but does not fully meet PLOS ONE’s publication criteria as it currently stands. Therefore, we invite you to submit a revised version of the manuscript that addresses the points raised during the review process.

We look forward to receiving your revised manuscript.

Kind regards,

Consolato M. Sergi

Academic Editor

PLOS ONE

Journal Requirements:

“..This work would not have been possible without the unwavering support of the St. Jude Global and University of North Carolina USA, which provided funding for utilization of these samples for research, and technical assistance respectively.”

Reviewers' comments:

Reviewer's Responses to Questions

**Comments to the Author**

1. Is the manuscript technically sound, and do the data support the conclusions?

Reviewer #1: Yes

Reviewer #2: Yes

2. Has the statistical analysis been performed appropriately and rigorously? 

Reviewer #1: Yes

Reviewer #2: N/A

3. Have the authors made all data underlying the findings in their manuscript fully available?

Reviewer #1: Yes

Reviewer #2: Yes

4. Is the manuscript presented in an intelligible fashion and written in standard English?

Reviewer #1: Yes

Reviewer #2: Yes

5. Review Comments to the Author

Reviewer #1: The authors have successfully utilized two frameworks, EPIS and ERIC, to examine the biobank implementation process in Pakistan. However, to enhance the clarity and impact of the conclusions, the overlooked aspects relevant to low- and middle-income countries should be explicitly addressed. Specifically, the authors could list the key areas requiring attention to support the development and sustainability of pediatric biobanking in Pakistan.

For instance, the following could be added as a penultimate sentence in the conclusion:

“To effectively apply the ERIC framework to pediatric cancer biobanking in Pakistan, the following aspects need further development: improving physical infrastructure, optimizing existing resources, establishing new communication channels, and implementing targeted training programs.”

Reviewer #2: This paper is well written. It addresses many potential challenges related to effective specimen management. However, in my opinion, ethical issues should be discussed more in detail. In Pakistan where a big part of population is in rural areas and may be uneducated, ethical and legal issues need to be looked at and discussed a bit more in detail.

6. PLOS authors have the option to publish the peer review history of their article (what does this mean? ). If published, this will include your full peer review and any attached files.

**Do you want your identity to be public for this peer review?** For information about this choice, including consent withdrawal, please see our Privacy Policy .

Reviewer #1: **Yes: ** Joseph F Feulefack

Reviewer #2: No

---

## [Author Response · Author response to Decision Letter 1]

13 Feb 2025

Thank you for your careful review of our manuscript titled “From blueprint to biobank: leveraging Expert Recommendations for Implementing Change (ERIC) to pediatric cancer biobanking in Pakistan”. We have tried to address all comments and suggestions. We believe the constructive feedback has allowed significant improvements to the manuscript. Below are our point-by-point responses.

Reviewer 1:

Comment: “The authors have successfully utilized two frameworks, EPIS and ERIC, to examine the biobank implementation process in Pakistan. However, to enhance the clarity and impact of the conclusions, the overlooked aspects relevant to low- and middle-income countries should be explicitly addressed. Specifically, the authors could list the key areas requiring attention to support the development and sustainability of pediatric biobanking in Pakistan.

For instance, the following could be added as a penultimate sentence in the conclusion:

“To effectively apply the ERIC framework to pediatric cancer biobanking in Pakistan, the following aspects need further development: improving physical infrastructure, optimizing existing resources, establishing new communication channels, and implementing targeted training programs.”

Response: Thank you for this suggestion. The following has been added as the concluding sentence:

“Further development of physical infrastructure, optimization of existing resources, and implementation of targeted training programs will be indispensable for further development and sustainability of pediatric cancer biobanking in Pakistan, as will be implementation of system-level strategies frequently overlooked in low- and middle-income countries, including addressing regulatory environments and community engagement.”

Reviewer 2:

Comment 1: “Do authors envision data sharing and collaboration at international level? If so, authors need to comment on how this model may deal with potential problems such as, data protection laws, and logistical issues.”

Response: Yes, data sharing and international collaborations are envisaged. Thank you for this observation. Accordingly, the following has been added to the first paragraph (page 20-21) of the section “Sustainment”.

“To enable international data sharing and collaborations, a process of developing material and data transfer agreements, compliant with the legal frameworks and ethical considerations is in place. Currently, data sharing is done following ethical and legal approval, as well as through utilizing secure cloud-based platforms which meet international data security standards for data transfer. As the biobank expands, standardized data formats and protocols compatible with major international biobank networks will need to be developed. Moreover, the consent process necessitates prior approval of deidentified data sharing.”

Comment 2: “Authors need to be more explicit about ethical and legal considerations. Authors have mentioned some related issues briefly. However, they have not discussed the challenges of building trust and engaging with the community and stakeholders, which is essential for the success of a biobank. In fact in a country of limited resources, the ethical issues could be enormous. There is a risk of exploiting vulnerable populations, such as those in desperate financial situations, by offering monetary compensation for plasma donations. This can lead to individuals donating more frequently than is safe, potentially compromising their health, maintaining high standards of safety and quality in plasma collection and storage can be challenging in resource-limited settings. This includes ensuring proper sterilization, handling, and storage to prevent contamination and degradation of plasma, Inadequate regulatory frameworks and enforcement can lead to unethical practices and compromised safety standards. Ensuring that plasma collection centers adhere to strict guidelines and are regularly inspected is essential. Authors may briefly address these points as well.”

“This paper is well written. It addresses many potential challenges related to effective specimen management. However, in my opinion, ethical issues should be discussed more in detail. In Pakistan where a big part of population is in rural areas and may be uneducated, ethical and legal issues need to be looked at and discussed a bit more in detail.”

Response: Thank you very much for highlighting this important aspect. We have tried to discuss these issues in more detail, in particular reference to our patient population. Several paragraphs have been added to address this aspect in the “Implementation” section (page 14-15). The relevant additions are as follows:

“The Belmont Principle of respect for persons requires particular consideration in these circumstances given the vulnerable populations involved, including children and individuals not formally educated. Ethical dilemmas can potentially arise with more than minimal risk research, given the unresolved debates on whether a large societal benefit can outweigh individual risk, and if such research should proceed at all. At the same time the principle of justice also demands equitable distribution of not just risks but also benefits. While sample selection is based purely on technical eligibility rather than social characteristics, the inclusion of vulnerable populations is necessary for developing targeted treatments that will benefit these communities.

To minimize any perceived coercion, consent is obtained by independent personnel unrelated to treatment. Staff clearly communicate that treatment decisions are entirely independent of participation. Notwithstanding, some degree of uncertainty about the completeness of relevant information transferred and comprehended currently remains inevitable for our patient population. Future work in this direction may entail extensive, close work with relevant communities through identification of individuals proficient in English, Urdu, and the local language, together with the right level of education and cultural understanding, to design non-written (e.g. audio) supplementation of consent form information.

Protocol safeguards for biological sample collection establish strict protective measures for donors. These include prohibiting monetary compensation to prevent exploitation and mandatory physician approval for collection to ensure samples are not drawn when it may not be absolutely safe for the patient. Collections are aligned with diagnostic draws to maintain standard safety protocols. While physical risks from sample collection are minimal, others including psychological, social, legal, and economic, are minimized, though not entirely eliminated through de-identification, and other measures to protect privacy and confidentiality.”

Comment 3: “Financial constraints, funding and effective education of patient/participant in this area are going to be a challenge to implement this strategy nationwide, especially in rural areas. In Pakistan where a majority of population resides in rural areas, practical implication of this strategy may be a challenge.”

Response: The following sentence has been added to the last paragraph (page 22) of the section “Sustainment”, where we have touched upon scale-up issues: “Additional challenges include effective education of participants in semi-urban or rural areas”

---

## [Decision Letter · Decision Letter 1]

4 Mar 2025

From Blueprint to Biobank: Leveraging Expert Recommendations for Implementing Change (ERIC) to Pediatric Cancer Biobanking in Pakistan

PONE-D-24-44873R1

Dear Dr. Aijaz,

We’re pleased to inform you that your manuscript has been judged scientifically suitable for publication and will be formally accepted for publication once it meets all outstanding technical requirements.

Kind regards,

Consolato M. Sergi

Academic Editor

PLOS ONE

Additional Editor Comments (optional):

Reviewers' comments:

Reviewer's Responses to Questions

**Comments to the Author**

1. If the authors have adequately addressed your comments raised in a previous round of review and you feel that this manuscript is now acceptable for publication, you may indicate that here to bypass the “Comments to the Author” section, enter your conflict of interest statement in the “Confidential to Editor” section, and submit your "Accept" recommendation.

Reviewer #1: All comments have been addressed

Reviewer #2: All comments have been addressed

2. Is the manuscript technically sound, and do the data support the conclusions?

Reviewer #1: Yes

Reviewer #2: Yes

3. Has the statistical analysis been performed appropriately and rigorously? 

Reviewer #1: Yes

Reviewer #2: Yes

4. Have the authors made all data underlying the findings in their manuscript fully available?

Reviewer #1: Yes

Reviewer #2: Yes

5. Is the manuscript presented in an intelligible fashion and written in standard English?

Reviewer #1: Yes

Reviewer #2: Yes

6. Review Comments to the Author

Reviewer #1: I thank the authors for their detailed responses to the suggestions made. All the requested changes have been completed.

Reviewer #2: (No Response)

7. PLOS authors have the option to publish the peer review history of their article (what does this mean? ). If published, this will include your full peer review and any attached files.

**Do you want your identity to be public for this peer review?** For information about this choice, including consent withdrawal, please see our Privacy Policy .

Reviewer #1: **Yes: ** Joseph Feulefack

Reviewer #2: No

---

## [Editor Report · Acceptance letter]

PONE-D-24-44873R1

PLOS ONE

Dear Dr. Aijaz,

I'm pleased to inform you that your manuscript has been deemed suitable for publication in PLOS ONE. Congratulations! Your manuscript is now being handed over to our production team.

Kind regards,

on behalf of

Professor Consolato M. Sergi

Academic Editor

PLOS ONE